Fungal communities represent the majority of root-specific transcripts in the transcriptomes of Agave plants grown in semiarid regions

Marone Marina Püpke 1
Campanari Maria Fernanda Zaneli 1
Raya Fabio Trigo 1
Pereira Gonçalo Amarante Guimarães goncalo@unicamp.br 1 2
Carazzolle Marcelo Falsarella 1 2
1 Department of Genetics, Evolution, Microbiology, and Immunology, University of Campinas , Campinas , São Paulo , Brazil
2 Center for Computing and Engineering Sciences, University of Campinas , Campinas , São Paulo , Brazil
LaMontagne Michael
Electronic publication date: 2022 May 2
Publication date: 2022
Volume: 10
Electronic Location ID: e13252
Received 2021 Sep 29; Accepted 2022 Mar 21
Copyright: ©2022 Marone et al.
Copyright year: 2022
Copyright holder: Marone et al.
License: This is an open access article distributed under the terms of the Creative Commons Attribution License, which permits unrestricted use, distribution, reproduction and adaptation in any medium and for any purpose provided that it is properly attributed. For attribution, the original author(s), title, publication source (PeerJ) and either DOI or URL of the article must be cited.
License URL: https://creativecommons.org/licenses/by/4.0/

Keywords: Agave, Microbiome, Transcriptome, Root, Drought resistance, Fungal communities

Funding: Coordenação de Aperfeiçoamento de Pessoal de Nível Superior - Brazil (CAPES) n. 88887.373979/2019-00 88887.479629/2020-00 Center for Computational Engineering and Sciences - FAPESP/Cepid 2013/08293-7 The São Paulo Research Foundation (FAPESP) 2020/02524-0 CNPq (Nexus Project: Integration Caatinga-Sisal ) n. 441625/2017-7 This work was supported by Coordenação de Aperfeiçoamento de Pessoal de Nível Superior - Brazil (CAPES) n. 88887.373979/2019-00 and 88887.479629/2020-00, Center for Computational Engineering and Sciences - FAPESP/Cepid (2013/08293-7), the São Paulo Research Foundation (FAPESP) grant 2020/02524-0, and CNPq (Nexus Project: Integration Caatinga-Sisal n. 441625/2017-7). The funders had no role in study design, data collection and analysis, decision to publish, or preparation of the manuscript.

==============================
Agave plants present drought resistance mechanisms, commercial applications, and potential for bioenergy production. Currently, Agave species are used to produce alcoholic beverages and sisal fibers in semi-arid regions, mainly in Mexico and Brazil. Because of their high productivities, low lignin content, and high shoot-to-root ratio, agaves show potential as biomass feedstock to bioenergy production in marginal areas. Plants host many microorganisms and understanding their metabolism can inform biotechnological purposes. Here, we identify and characterize fungal transcripts found in three fiber-producing agave cultivars (Agave fourcroydes, A. sisalana, and hybrid 11648). We used leaf, stem, and root samples collected from the agave germplasm bank located in the state of Paraiba, in the Brazilian semiarid region, which has faced irregular precipitation periods. We used data from a de novo assembled transcriptome assembly (all tissues together). Regardless of the cultivar, around 10% of the transcripts mapped to fungi. Surprisingly, most root-specific transcripts were fungal (58%); of these around 64% were identified as Ascomycota and 28% as Basidiomycota in the three communities. Transcripts that code for heat shock proteins (HSPs) and enzymes involved in transport across the membrane in Ascomycota and Basidiomycota, abounded in libraries generated from the three cultivars. Indeed, among the most expressed transcripts, many were annotated as HSPs, which appear involved in abiotic stress resistance. Most HSPs expressed by Ascomycota are small HSPs, highly related to dealing with temperature stresses. Also, some KEGG pathways suggest interaction with the roots, related to transport to outside the cell, such as exosome (present in the three Ascomycota communities) and membrane trafficking, which were further investigated. We also found chitinases among secreted CAZymes, that can be related to pathogen control. We anticipate that our results can provide a starting point to the study of the potential uses of agaves’ fungi as biotechnological tools.

Introduction

Plants host many microorganisms that have crucial roles in plant development, growth, adaptability, and diversity (Trivedi et al., 2020). Plant microbiomes are composed of endophytes—which colonize the tissues and occupy the intra and intercellular spaces in at least one period of their life cycle—and epiphytes—which colonize the vegetal surface (Fitzpatrick et al., 2020). Understanding these microbial communities’ structures and their interaction mechanisms with plants can lead to phenotypes of interest, such as drought resistance and plant growth promoters, assisting in higher productivities (Wani et al., 2015).

Agaves are drought-resistant semiarid plants with commercial uses and potential to be used as feedstock for bioenergy production in marginal areas. Currently, agaves are only used to produce sisal fibers (mainly in Brazil) and alcoholic beverages, such as tequila and mezcal (in Mexico). However, agaves also show promise as feedstock for bioenergy production in marginal areas because of high productivities (Owen, Fahy & Griffiths, 2016), low lignin content (Yang et al., 2015), and high shoot-to-root ratio (Borland et al., 2009). Agaves also have several drought resistance mechanisms, such as the crassulacean acid metabolism (CAM), the most water-use efficient photosynthesis (Borland et al., 2009). Additionally, they possess retractable roots (Blunden, Yi & Jewers, 1973), waxy epidermis, and sunken stomata (Davis & Long, 2015) to avoid water loss.

Agave’s morphological and physiological adaptations to dry climates have been more studied than other aspects, such as molecular mechanisms and genetics. Some recent research has focused on the molecular aspects of agave’s CAM metabolism and drought resistance mechanisms, mainly with the species A. tequilana Weber var. azul and A. americana (Gross et al., 2013; Yang et al., 2015; Abraham et al., 2016). Considering fiber-producing agave cultivars, such as A. fourcroydes, A. sisalana, and hybrid 11648 ((A. angustifolia x A. amaniensis) x A. amaniensis), there are just a few recent studies available (Huang et al., 2018; Huang et al., 2019; Sarwar et al., 2019; Raya et al., 2021), mostly approaching plant physiological mechanisms related to drought or cell wall biosynthesis. However, one aspect still to be vastly explored is the microbiome.

In agave plants, tolerance to heat and drought stress partly might be due to the microorganisms that inhabit them. Indeed, a study found the mycobiota from A. tequilana and A. salmiana produced volatile organic compounds that improved plant growth (Camarena-Pozos et al., 2021). Thus, knowing these microorganisms and their possible impacts on plants becomes relevant. In agave, classical protocols for isolation and/or inoculation of growth promoting microorganisms were performed (Pimienta-Barrios, Zanudo-Hernandez & Lopez-Alcocer, 2009; Ruiz et al., 2013; Martínez-Rodríguez et al., 2014; Quinones-Aguilar et al., 2016; Montoya Martinez et al., 2019), and more recently metagenomics and metatranscriptomics approaches were applied (Coleman-Derr et al., 2016; Citlali et al., 2018; Flores-Núñez et al., 2020). In Brazil, three studies explored the microbial community of Agave sisalana but used only classical microbiology methods to isolate endophytic fungi and prokaryotes (Candeias et al. 2016; Damasceno et al., 2019).

Metatranscriptomics can identify genes expressed in microbiomes and, therefore, possible plant-microbe interactions. However, there are some challenges in metatranscriptomics such as dependence on the availability of information in databases for taxonomic characterization (Shakya, Lo & Chain, 2019) and complications in RNA extraction depending on the type of sample (e.g., soil) (Mukherjee & Reddy, 2020). Furthermore, most meta-omics tools are better applied for prokaryotic organisms (Shakya, Lo & Chain, 2019).

In our previous work (Raya et al., 2021), we assembled and analyzed the comprehensive transcriptomes (leaf, stem, and root tissues) of three fiber-producing agave cultivars collected at noon in a germplasm bank located in Monteiro, Paraíba, Brazil. This region faced an irregular rainfall regime before sampling. Two of the cultivars are the most used in fiber production in Brazil—A. sisalana and hybrid 11648—and one is the most used in Mexico—A. fourcroydes.

In this paper, we used bioinformatic analyses to identify and characterize fungal communities found on three agave transcriptomes. We (1) assessed the number of fungal transcripts, their location in the plant tissues, and their expression values; (2) assessed their taxonomic affiliations; (3) performed functional annotation, focusing on transporters and carbohydrate-degrading enzymes (CAZymes), and (4) investigated potential heat shock proteins.

Methods

Transcriptome sequencing, assembly, and quantification

Samples of A. fourcroydes, A. sisalana, and hybrid 11648 were collected at the city of Monteiro, state of Paraíba, Brazil, at the agave germplasm bank owned by Embrapa (Brazilian Agricultural Research Corporation). The germplasm bank is at a semi-arid climate area with non-calcic brown soil. Before sampling, the precipitation regime was highly irregular (Fig. S1); sampling occurred on July 6th, 2016, and the last precipitation happened on May 31st, 2016. We harvested, at noon, three different individuals (apparently healthy), for each cultivar. Plants were around 7 years old and we collected root, stem, and leaf samples. Root samples were collected from 0–10 cm depths. Samples were not disinfected, so both epiphytic and endophytic microorganisms might be present. Total RNA was obtained with the protocol proposed by Zeng & Yang (2002) with modifications by Le Provost et al. (2003). Libraries were prepared with 1 μg total RNA using the KAPA Stranded mRNA-Seq kit, following the manufacturer’s protocol, isolating only poly(A) tails. Sequencing was performed using Illumina/HiSeq 4000, generating nine libraries for each cultivar with millions of 50bp paired-end reads. Raw data is available in NCBI SRA (accession number PRJNA746623).

Reads were assembled into three separate transcriptomes for each agave cultivar using Trinity v. 2.5.1 (Grabherr et al., 2011). We used Transdecoder v. 5.0.2 (Haas et al., 2013) for ORF prediction, configured to a minimum length of 200 nucleotides. We performed an additional step to select the longest isoform for each locus, considering only those with TPM (Transcript per Million) values greater than 1 and ORF length greater than 255 nucleotides. Expression values were obtained with kallisto v. 0.44.0 (Bray et al., 2016) with 100 bootstraps, returning TPM values. For all further analysis, the mean expression value from the three replicates was used. For each transcriptome, we performed tissue specificity analysis using the tspex program (Camargo et al., 2020) considering a threshold of SPM > 0.95 to identify tissue-specific transcripts. Annotation of the whole transcriptome was performed with PANNZER2 (Törönen, Medlar & Holm, 2018) and remaining unannotated sequences were submitted to a BLASTp against Uniref90 (E-value 1e−5). During the annotation step we identified that along with plant transcripts, many fungal transcripts were assembled, which led us to develop a new pipeline for their correct identification and further analysis.

In-house pipeline to separate plant and fungal transcripts

We used two approaches to extract fungal transcripts from the three plant transcriptomes. Kaiju software v. 1.6.3 (Menzel, Ng & Krogh, 2016) was used to obtain the fungal sequences and to infer the fungal taxonomic classification by comparison with the NCBI/NR database with an E-value threshold of 1e−5. From this classification, count matrices were generated in the format of reads per taxon, which were compared at the phylum and genus levels. Sequences annotated as plants (higher level: Streptophyta division) were excluded. However, manual annotation revealed that many plant transcripts with very high expression were mistakenly classified as fungal sequences, so we decided to use another approach as well.

The second approach was based on the similarities and differences between the plant and fungal protein sequences available in the Uniref90 database. For this, we performed a BLASTx (E-value threshold of 1e−10) of the assembled transcriptome of A. fourcroydes, A. sisalana, and hybrid 11648 against Uniref90 and selected all transcripts that presented at least 70% of the top 10 hits identified as fungi in Taxonomy DB. Only the results from the second approach were considered for subsequent analyses.

Taxonomic classification of the fungal transcripts

Taxonomic classification from the software Kaiju was used as a guide to perform a pipeline to separate the main fungal phyla (Ascomycota and Basidiomycota) in our datasets. We developed a Perl pipeline (Data S1) to test different parameters to optimize this classification. From 1,014 fungal genomes available on Ensembl Fungi release 46 (https://fungi.ensembl.org), we randomly selected 2,000 CDS for Ascomycota and 700 for Basidiomycota to be used as a training and test dataset. These proportions were based on the preliminary analysis with Kaiju software which showed that the fungal transcripts are mostly represented by Ascomycota and Basidiomycota species. Because there were many parameters to consider, being a multivariate set, we had to test the sensitivity of each parameter after fixating one of them.

Thus, we used similarities in nucleotides (BLASTn) and amino acids (tBLASTx) to evaluate the classification capacity (using metrics TPR - true positive rate - and FPR - false positive rate) between the groups “Ascomycota”, “Basidiomycota”, “Asco or Basidio” or “Other Fungi”. These 2,700 CDSs were blasted against all CDS from 1,014 fungal genomes (after the exclusion of these 2,700 CDS to avoid self-identification). The parameters of E-value threshold (1e−10 and 1e−20) and the minimum number N of top BLAST hits (top N BLAST hits) were optimized. If a transcript could not be distinguished between Ascomycota and Basidiomycota based on our criteria, it would be classified as “Asco or Basidio”. Moreover, if the transcript did not present similarity with Ascomycota nor Basidiomycota, but presented similarity with other fungal genomes, it was assigned to “Other Fungi”.

Orthologous gene analysis

To compare the three fungal communities’ nucleotide sequences, an orthologous analysis was done with software OrthoFinder v. 2.5.2 (Emms & Kelly, 2019) configured with the parameter “-d” (nucleotide similarity). Then, we counted the number of orthologous nucleotide families exclusive to (orphan genes) or shared between Ascomycota and Basidiomycota.

Ribosomal RNA identification

Ribosomal RNA identification was performed using the full set of raw sequence reads. We used SortMeRNA (Kopylova, Noé & Touzet, 2012) with default parameters to filter the reads that had similarities to the 28S rRNA database. Then we filtered the reads belonging to fungi to assess percentages of fungal reads in our datasets. Finally, we listed the top 5 fungal species that had more abundant reads in each RNA-seq library.

Functional annotation of fungal transcripts and enrichment tests

Fungal transcripts from the three plants were further annotated with RPS-Blast (E-value threshold of 0.01) using the CDD database (Lu et al., 2020) as reference. The EggNOG-mapper program was used (Huerta-Cepas et al., 2019) to identify KEGG pathway (KO) groups (Kanehisa et al., 2019). Using an R script (Data S1), each fungal transcript was grouped considering the annotation of CDD and KO and divided between “Ascomycota”, “Basidiomycota”, “Asco or Basidio”, and “Other Fungi” groups. Significant KEGG terms and CDDs were detected by the hypergeometric test using all fungal transcripts as background, being accepted those with p-value < 0.05.

Transport proteins, CAZymes, and secreted proteins prediction

Membrane transport proteins were identified using BLASTp against the curated Transporter Classification Database (TCDB) (Saier et al., 2016) with a threshold E-value of 1e−5 and an alignment coverage of at least 70%. To identify carbohydrate-active enzymes (CAZymes), we used the dbCAN software (Yin et al., 2012). We also used SignalP v. 5.0 (Almagro Armenteros et al., 2019) to identify sequences of signal peptides in all fungal transcripts, with a threshold of score >0.5 for secreted proteins.

Heat shock protein orthologous analysis

We compared heat shock proteins (HSPs) between our six fungal datasets (Ascomycota and Basidiomycota sequences from A. fourcroydes, A. sisalana, and hybrid 11648) and all Ascomycota and Basidiomycota available genomes in Ensembl Fungi release 46. To do so, we searched for HSP domains using the manually curated database HSPIR (Ratheesh Kumar et al., 2012), which has sequences from the six major groups of HSPs. To obtain all heat shock proteins in our fungal datasets and the fungal genomes (1,014 genomes), we searched with HMMER (Eddy, 2008) with an E-value threshold of 0.001. Then, we selected the top 10 genomes for Ascomycota and the top 10 for Basidiomycota that had more members of HSP families (see the distribution in Fig. S2). Using only the HSPs sequences from these 20 datasets and our fungal community datasets divided into Ascomycota and Basidiomycota for each cultivar, we ran a protein orthologous analysis using OrthoFinder (Emms & Kelly, 2019).

Assignment of transcripts to the genus Talaromyces

We did a BLASTn (E-value < 1e−10) of all Ascomycota transcripts against a set of concatenated genomes of Talaromyces available at Ensembl Fungi release 46. We selected only blast hits with alignment coverage above 70%.

Results

Root-specific fungi in Agave plants

Transcriptomic analysis of three fiber-producing agave cultivars (A. fourcroydes, A. sisalana, hybrid 11648) from leaf, stem, and root tissues revealed a large number of fungal transcripts (Table 1). Among the three agave transcriptomes, there was an average of 12% fungal transcripts, which totaled 2,966, 4,313, 3,433 from A. fourcroydes, A. sisalana, and hybrid 11648, respectively. Although these transcripts represented a high percentage within the complete dataset, they did not present very high expression values, as these were calculated based on abundances of the plant transcripts (average of the three datasets top expressed plant transcript is 27,288 TPM); the highest fungal transcripts expression values were 197, 181 and 413 TPM for A. fourcroydes, A. sisalana and hybrid 11648, respectively (Table S1). Fungal transcript and protein sequences for each cultivar are available in Data S2.

Table 1 Summary of assembly parameters.

Plant transcriptome assemblies were performed with Trinity and ORF prediction was performed using Transdecoder (minimum length of 200 nucleotides). All transcripts were submitted to a BLASTx against Uniref90 for fungal transcript assignment. Transcripts with at least 70% of the top 10 hits identified as fungi in the Taxonomy DB were considered fungal ones.

Parameter	A. fourcroydes	A. sisalana	Hybrid 11648	
Number of fungal transcripts	2,996	4,313	3,433	
Mean length (bp)	545	540	529	
N50	579	570	552	
Max/min CDS length (bp)	3,099/258	3,210/255	2,649/264	

Fungal transcripts were largely root-specific (i.e., specificity measure, SPM > 0.95). Root-specific fungal transcripts were more abundant than root-specific plant transcripts in the three agave transcriptomes (Fig. 1A) (around 58% of all root-specific transcripts belong to fungi). Nevertheless, there are six, three and nine expressed transcripts in other tissues (for A. fourcroydes, A. sisalana, and hybrid 11648, respectively). These transcripts present high expression values in two or three tissues (Table S1). Among them, there are some annotated as heat shock proteins. Indeed, the most expressed transcript in hybrid 11648 was annotated as Hsp70 and has high expression values in the three tissues (leaf = 222, stem = 85, and root = 413 TPM).

Figure 1 General numbers of fungal and plant transcripts in each plant and summary of the orthologous analysis of the three fungal communities.

(A) The plant transcriptomes were assembled de novo. Among the assembled transcripts we found fungal transcripts, which are almost exclusively expressed in the plants’ roots. Indeed, most root-specific transcripts were fungal. (B) Number of orthologous gene families (nucleotides) in each fungal community. Numbers in square brackets show only Ascomycota annotated families and in curled brackets are Basidiomycota annotated families. Photos: Fabio T. Raya.

Nucleotide orthologous analysis between the three communities showed they share a very similar core of expressed transcripts, only presenting a few exclusive gene families (Fig. 1B; Table S2). As expected, due to the greater amount of Ascomycota transcripts, we found a higher proportion of Ascomycota-exclusive families (square brackets) compared to Basidiomycota (curled brackets); however, the opposite occurs in hybrid 11648-exclusive families (five and 18 for Ascomycota and Basidiomycota, respectively). Furthermore, we found that more than half of all transcripts are orphans (1,484, 2,702, and 1,619 for A. fourcroydes, A. sisalana, and hybrid 11648, respectively), i.e., don’t form orthologous clusters with any other gene.

Taxonomic analysis

Taxonomy inference of the whole transcriptomes using the Kaiju software classified 12% ± 1.5 (values are the mean for the three plant communities) as fungal transcripts, of which 68% ± 5.5 were Ascomycota and 30% ± 5.3 Basidiomycota. The most represented Ascomycota genera were Talaromyces (9% ± 1.5) and Corynespora (4% ± 0.2). For Basidiomycota, there was resolution only to classify until order, of which the main were Agaricales (6% ± 2.5) and Auriculariales (3% ± 1.4). However, this approach misclassified many plant transcripts as fungal ones, so we used a Perl pipeline to classify transcripts between Ascomycota and Basidiomycota, as these were the most abundant phyla (information obtained with Kaiju).

This new approach was based on sequence similarity using 1,014 public fungal genomes and their respective coding sequences (more details in the Methods section). Results of the parameter optimization step are on Data S3. After parameter optimization, we applied the pipeline to classify the fungal transcripts of the three cultivars into the “Ascomycota”, “Basidiomycota”, “Asco or Basidio”, and “Other Fungi” groups (Table 2; Table S1).

Table 2 Transcript numbers in each fungal group.

Transcripts were identified with our in-house pipeline described in the Methods section. “Asco or Basidio” refers to transcripts that were either Ascomycota or Basidiomycota but could not be classified.

Cultivar	Ascomycota	Basidiomycota	Asco or Basidio	Other fungi	Total	
Agave fourcroydes	1,927	797	19	253	2,996	
Agave sisalana	3,012	1,036	23	242	4,313	
Hybrid 11648	1,986	1,179	18	250	3,433	

Ribosomal RNA analysis

To confirm the previous results, we returned to the raw reads and searched the rRNA fungal dataset with SortMeRNA. The rRNA analysis confirmed that root samples presented much higher percentages of fungal rRNA (between 6% and 20%) than leaf and stem (below 1%) (Table S3). The most abundant groups in the roots were Parastagonospora, Talaromyces, and Cryptococcus, while in the other tissues the most abundant were Cryptococcus and Verrucaria. We could also confirm that Ascomycota and Basidiomycota are the most represented groups and that the Talaromyces genus is in the top five abundant groups.

Functional annotation of the fungal transcripts

Functional annotation analysis is shown in Fig. 2 as frequencies of statistically significant (p-value < 0.05) enriched KEGG pathways (Figs. 2A and 2C) and conserved domains (CDD) (Figs. 2B and 2D). Generally, Ascomycota presented more categories than Basidiomycota, as Ascomycota has more transcripts.

Figure 2 Functional characterization of KEGG pathways and protein domains.

(A, C) Frequency of enriched KEGG pathways for (A) Ascomycota and (C) Basidiomycota. (B, D) Frequency of conserved protein domains (CDD) for (B) Ascomycota and (D) Basidiomycota. The hypergeometric test was used with p-value < 0.05 and only significantly enriched terms are shown.

The most frequent enriched KEGG pathway in both Ascomycota and Basidiomycota was related to chaperones and folding catalysts, absent only in the A. fourcroydes Basidiomycota community. Some pathways suggest interaction with the roots, related to transport to outside the cell, such as exosome (present in the three Ascomycota communities, but exclusive in Basidiomycota in A. sisalana), membrane trafficking, and transporters. Other routes could be associated with root development and elongation, such as Citrate Cycle (TCA) and Glyoxylate and dicarboxylate metabolism, observed exclusively in the samples of A. fourcroydes for Basidiomycota. Still regarding metabolism, Glycosylphosphatidylinositol (GPI) is present in all three datasets of Ascomycota and GTP-binding protein is exclusive of Basidiomycota in A. sisalana.

Table 3 Top expressed transcripts identified as transporters in fungal communities of the three agave cultivars.

Protein sequences were blasted against the Transporter Classification Database (TCDB) with E-value < 1e–5 and filtered for alignment coverage > = 70%. Expression values are in TPM.

Cultivar	ID	TCDB family	Fungal classification	Root mean expression (TPM)	
A. fourcroydes	AF_DN51128_c6_g2	The HSP90/CDC37 (HSP90/CDC37)	Ascomycota	121.53	
AF_DN37348_c0_g1	The Endoplasmic Reticular Retrotranslocon (ER-RT)	Basidiomycota	97.41	
AF_DN51128_c6_g1	The HSP90/CDC37 (HSP90/CDC37)	Basidiomycota	86.05	
AF_DN46089_c1_g1	The Mitochondrial Carrier (MC)	Basidiomycota	69.20	
AF_DN39498_c0_g1	The Endoplasmic Reticular Retrotranslocon (ER-RT)	Asco or Basidio	63.63	
AF_DN52052_c2_g1	The Endoplasmic Reticular Retrotranslocon (ER-RT)	Ascomycota	48.26	
AF_DN103552_c0_g1	The Cation Channel-forming Heat Shock Protein-70 (Hsp70)	Ascomycota	48.24	
AF_DN42546_c0_g1	The Cation Channel-forming Heat Shock Protein-70 (Hsp70)	Basidiomycota	36.56	
AF_DN44188_c2_g2	The Cation Channel-forming Heat Shock Protein-70 (Hsp70)	Ascomycota	28.82	
AF_DN50952_c2_g1	The Nuclear mRNA Exporter (mRNA-E)	Ascomycota	27.95	
A. sisalana	AS_DN53864_c3_g1	The HSP90/CDC37 (HSP90/CDC37)	Ascomycota	99.52	
AS_DN59592_c8_g2	The Endoplasmic Reticular Retrotranslocon (ER-RT)	Ascomycota	99.50	
AS_DN51359_c2_g1	The Endoplasmic Reticular Retrotranslocon (ER-RT)	Ascomycota	71.33	
AS_DN53419_c0_g2	The Cation Channel-forming Heat Shock Protein-70 (Hsp70)	Ascomycota	67.31	
AS_DN53419_c0_g3	The Cation Channel-forming Heat Shock Protein-70 (Hsp70)	Ascomycota	56.72	
AS_DN50667_c0_g1	The Mitochondrial Carrier (MC)	Ascomycota	47.87	
AS_DN54411_c1_g1	The Mitochondrial Carrier (MC)	Ascomycota	28.98	
AS_DN30106_c0_g1	The Cation Channel-forming Heat Shock Protein-70 (Hsp70)	Ascomycota	24.04	
AS_DN48395_c0_g1	The Cation Channel-forming Heat Shock Protein-70 (Hsp70)	Basidiomycota	23.72	
AS_DN56192_c3_g1	The Nuclear mRNA Exporter (mRNA-E)	Ascomycota	21.97	
Hybrid 11648	HY_DN39331_c2_g1	The Endoplasmic Reticular Retrotranslocon (ER-RT)	Ascomycota	413.13	
HY_DN32985_c0_g1	The Mitochondrial Carrier (MC)	Basidiomycota	46.37	
HY_DN36716_c2_g1	The Endoplasmic Reticular Retrotranslocon (ER-RT)	Ascomycota	37.86	
HY_DN36716_c3_g1	The Endoplasmic Reticular Retrotranslocon (ER-RT)	Ascomycota	32.70	
HY_DN28452_c0_g1	The Endoplasmic Reticular Retrotranslocon (ER-RT)	Basidiomycota	26.55	
HY_DN38958_c4_g2	The Cation Channel-forming Heat Shock Protein-70 (Hsp70)	Basidiomycota	23.08	
HY_DN39005_c7_g1	The HSP90/CDC37 (HSP90/CDC37)	Basidiomycota	22.64	
HY_DN10827_c0_g1	The Endoplasmic Reticular Retrotranslocon (ER-RT)	Ascomycota	21.63	
HY_DN70040_c0_g1	The Cation Channel-forming Heat Shock Protein-70 (Hsp70)	Ascomycota	20.58	
HY_DN39331_c3_g1	The Endoplasmic Reticular Retrotranslocon (ER-RT)	Ascomycota	19.66	

In regard to CDD, there were also domains related to chaperones and heat shock proteins. The most frequent domain present in all communities was ACD sHSPs-like (CDD:107221), a subunit of small heat shock proteins (HSP), that plays an important role in stress protection, and is found in prokaryotes and eukaryotes alike (Ganea, 2001). Some domains were enriched in just one community, such as “DnaJ” (CDD:199909) in hybrid 11648 in Ascomycota, which is also an HSP domain. For Basidiomycota, “HSP90” (CDD:333906) was exclusive in A. fourcroydes and “molecular chaperone DnaK” (CDD:234715) in hybrid 11648. Regarding primary metabolism, in the Ascomycota communities of A. sisalana and hybrid 11648 we observed “mannitol dehydrogenase (MDH)-like” (CDD:187610), which is responsible for catalyzing the conversion of fructose to mannitol. We also found the domain “Fungal hexose transporter”, which is specific to the hybrid 11648 Basidiomycota community.

Identification of transport proteins and carbohydrate-degrading enzymes

To deeper investigate transporters, we blasted our fungal transcripts against the curated TCDB. We found 145, 201, and 157 proteins related to transport with alignment coverage >70% in A. fourcroydes, A. sisalana, and hybrid 11648, respectively (Table S4). The profile of most frequent families is similar between the three communities (Table 3), although “Major Facilitator Superfamily (MFS)” only appears in Ascomycota of A. sisalana, and “H+- or Na+-translocating F-type, V-type and A-type ATPase (F-ATPase) Superfamily” in Basidiomycota of hybrid 11648. When looking at the most expressed transcripts annotated as transporters, there are two families related to heat shock proteins (“HSP90/CDC37 (HSP90/CDC37) Family” and “Cation Channel-forming Heat Shock Protein-70 (Hsp70) Family”). In tumor cells, members of the Hsp70 and Hsp90 families can be found in association with membranes, along with co-chaperones, regulating functions related to folding and trafficking (Gross et al., 2003; Heider et al., 2021). Another family with many highly expressed transcripts in all communities is “Endoplasmic Reticular Retrotranslocon (ER-RT)”, which is related to transport to and from the endoplasmic reticulum, mostly for degradation of misfolded proteins (Römisch, 2005). Overall, both Ascomycota and Basidiomycota presented highly expressed transporters (Table 2), except for A. sisalana, with the majority being from Ascomycota. Also, many families were related to regular transportation inside the cell, such as ABC transporters, ATPases, and transport to and from the mitochondria, and most of these have overall low expressions. More interestingly, we found transporters of ammonium (Q8NKD5—1.A.11.3.3) in A. fourcroydes and A. sisalana, phosphate (K4HTY2—2.A.1.9.11) in A. fourcroydes, and inorganic phosphate (Q7RVX9—2.A.1.9.2) in A. sisalana and hybrid 11648 with expressions varying from 0.84 to 5.56 TPM.

Carbohydrate Active enZYmes (CAZymes) were identified among all fungal transcripts and compared between Ascomycota and Basidiomycota (Fig. 3A). The majority were classified as glycoside hydrolases (GH) and the profile between Ascomycota and Basidiomycota in each plant is different, although the pattern is similar when comparing the same phyla. Considering percentages, Ascomycota presented more GHs than Basidiomycota, but the latter had more enzymes with auxiliary activities (AA), and polysaccharide lyases (PL) were exclusive to them. The majority of these CAZymes can be related to the fungi’s own carbohydrate metabolism.

Figure 3 Number of proteins identified as different classes of Carbohydrate Active enZYmes (CAZymes).

(A) All CAZymes found in each fungal community, showing the differences between the pattern of Ascomycota and Basidiomycota. (B) Comparison of secreted CAZymes between Ascomycota, Basidiomycota, and in the host plant. Secreted CAZymes have a signal peptide identified by SignalP. Asco, Ascomycota; Basidio, Basidiomycota; AA, auxiliary activities; CBM, carbohydrate-binding molecule; CE, carbohydrate esterases; GH, glycoside hydrolases; GT, glycosyltransferases; PL, polysaccharide lyases; AF, Agave fourcroydes; AS, Agave sisalana; HY, hybrid 11648.

To check for CAZymes that might be related to fungal-plant interactions, we compared secreted CAZymes of the three fungal communities to the ones secreted by the plant roots, considering root-specific transcripts (Fig. 3B). For Ascomycota, only A. sisalana presented types of CAZymes other than GHs. The general pattern between the three plants does not have a lot of variation except for A. sisalana, which presents more GHs. Focusing only on the fungal secreted CAZymes (Table 4), Basidiomycota secreted more CAZymes than Ascomycota, and GH128 is exclusive of Basidiomycota. Interestingly, most secreted CAZymes can be related to either plant cell wall degradation (GH10, GH11, GH16, GH17, GH43, AA9, and CE1) or fungal cell wall degradation (GH18 and GH128). Furthermore, hybrid 11648 presented GH10 and GH11 exclusively, which are related to the degradation of hemicellulose.

Heat shock protein orthologous analysis

Chaperones and heat shock proteins (HSPs) were one of the categories and pathways enriched and most frequent in the functional analysis, so we decided to carry out an analysis comparing our fungal communities with other fungal genomes. In our data, HSPs represented 6.21, 4.92, and 5.71% (5.61% average) of the total fungal transcripts for A. fourcroydes, A. sisalana, hybrid 11648, respectively. The total amount of transcripts annotated as HSPs by the HMM search was 433 Ascomycota and 161 Basidiomycota, in the three communities.

Table 4 All secreted CAZymes in the three fungal communities.

Plant cell wall (PCW) putative substrate was obtained from Kameshwar, Ramos & Qin (2019). Expression values are in TPM.

Cultivar	ID	CAZy ID	CAZy classification	PCW putative substrate	Fungal classification	Root mean expression (TPM)	
A. fourcroydes	AF_DN43332_c0_g1	GH79	β-glucuronidase	–	Asco or Basidio	6.92	
AF_DN16270_c0_g1	GH43	β-xylosidase	Hemicellulose	Basidiomycota	1.96	
AF_DN37792_c0_g1	GH18	chitinase	–	Basidiomycota	9.53	
AF_DN37367_c0_g1	GH18	chitinase	–	Basidiomycota	8.13	
AF_DN64727_c0_g1	GH16	β-glucanase	Hemicellulose	Ascomycota	2.52	
AF_DN54652_c0_g1	GH16	β-glucanase	Hemicellulose	Ascomycota	3.46	
AF_DN5518_c0_g1	GH128	endo-β-1,3-glucanase	–	Basidiomycota	7.91	
AF_DN44069_c0_g1	GH128	endo-β-1,3-glucanase	–	Basidiomycota	6.04	
AF_DN35139_c0_g1	GH128	endo-β-1,3-glucanase	–	Basidiomycota	7.18	
AF_DN84320_c0_g1	GH128	endo-β-1,3-glucanase	–	Basidiomycota	2.9	
AF_DN32526_c0_g1	AA9	Lytic cellulose monooxygenase	Cellulose/Hemicellulose	Basidiomycota	2.66	
AF_DN64239_c0_g1	AA9	Lytic cellulose monooxygenase	Cellulose/Hemicellulose	Basidiomycota	3.25	
A. sisalana	AS_DN131388_c0_g1	GH76	cell wall α-1,6-mannotransglycosylase/ α-1,6-mannanase	–	Basidiomycota	3.12	
AS_DN52648_c0_g1	GH76	cell wall α-1,6-mannotransglycosylase / α-1,6-mannanase	–	Ascomycota	4.39	
AS_DN25360_c0_g1	GH18	chitinase	–	Basidiomycota	9.34	
AS_DN46396_c2_g1	GH17	β-1,3-glucanase	Cellulose	Ascomycota	6.51	
AS_DN61972_c0_g1	GH17	β-1,3-glucanase	Cellulose	Ascomycota	1.65	
AS_DN41449_c0_g1	GH16	β-glucanase	Hemicellulose	Ascomycota	3.66	
AS_DN87265_c0_g1	GH16	β-glucanase	Hemicellulose	Basidiomycota	3.47	
AS_DN100681_c0_g1	GH128	endo-β-1,3-glucanase	–	Basidiomycota	12.16	
AS_DN42067_c0_g1	AA9	Lytic cellulose monooxygenase	Cellulose/Hemicellulose	Basidiomycota	2.31	
AS_DN114687_c0_g1	AA9	Lytic cellulose monooxygenase	Cellulose/Hemicellulose	Basidiomycota	3.5	
AS_DN40194_c0_g1	AA9	Lytic cellulose monooxygenase	Cellulose/Hemicellulose	Basidiomycota	5.42	
AS_DN16467_c0_g1	AA9	Lytic cellulose monooxygenase	Cellulose/Hemicellulose	Ascomycota	2.15	
Hybrid 11648	HY_DN20444_c0_g1	GH79	β-glucuronidase	–	Asco or Basidio	3.19	
HY_DN23374_c0_g1	GH76	cell wall α-1,6-mannotransglycosylase/ α-1,6-mannanase	–	Ascomycota	2.4	
HY_DN94917_c0_g1	GH18	chitinase	–	Basidiomycota	5.42	
HY_DN80813_c0_g1	GH17	β-1,3-glucanase	Cellulose	Ascomycota	1.27	
HY_DN39599_c0_g1	GH16	β-glucanase	Hemicellulose	Ascomycota	2.28	
HY_DN40185_c0_g1	GH128	endo-β-1,3-glucanase	–	Basidiomycota	4.89	
HY_DN27892_c0_g1	GH128	endo-β-1,3-glucanase	–	Basidiomycota	6.75	
HY_DN26068_c0_g1	GH11	Xylanase	Hemicellulose	Basidiomycota	6.82	
HY_DN56898_c0_g1	GH10	Xylanase	Hemicellulose	Ascomycota	5.87	
HY_DN50387_c0_g1	CE1	acetyl xylan esterase	Lignin	Basidiomycota	3.04	

To compare these numbers to other fungi, we did a protein orthologous analysis with HSPs prospected in 1,014 fungal genomes (Ensembl Fungi) and selected the top 10 genomes for Ascomycota and the top 10 for Basidiomycota with more HSPs. The list of these fungi can be found in Fig. 4. On the orthologous analysis, there were 165 HSP orthologous families (Table S5). Among these, only 22 contained at least one protein from our agave datasets (Fig. 4), of which 4 are exclusive (no orthologous in the other fungi’s genomes), presenting a minimum coverage of 70% compared to the expected length of the HSP type. These exclusive families were all annotated as ACD (alpha-crystallin domain), a domain present in small HSPs. Also, the family with more proteins from our datasets (OG0000001), which are more abundant in Ascomycota (a mean of 50 transcripts in the three sets), was annotated as ACD. The family with more proteins (419), considering all datasets, was annotated as DnaJ (OG000000), a type of co-chaperone that acts helping the folding performed by Hsp70 (Genest, Wickner & Doyle, 2019), although it is not abundant in our agave datasets, especially in Basidiomycota.

Figure 4 Protein orthologous analysis of heat shock proteins (HSPs) in the agave datasets (Ascomycota and Basidiomycota) and in 20 fungal genomes with more HSPs.

Based on the HMM search against the HSPIRDB, 10 Ascomycota and 10 Basidiomycota genomes with more HSPs were selected. Only families presenting at least one protein from the agave datasets are represented. Annotation was according to HSPIRDB. Scale is in number of proteins. AF, Agave fourcroydes; AS, Agave sisalana; HY, hybrid 11648.

The number of transcripts in each family varied little between the six analyzed communities, showing the similarity between the expressed major HSP groups. However, regarding expression values, these numbers varied a lot inside the families, with only a few transcripts with expression above 10 TPM and the vast majority (83.9%) with values between 0.77 and 9.87 TPM. The families with more highly expressed transcripts are OG0000001, OG0000002, OG0000008, OG0000030, and OG0000114 (Table S5).

We also analyzed all HSPs from the fungal genomes, as we did not find any studies performing such comparison. Considering all fungi species analyzed, the types of HSP with more protein orthologous families were Hsp100, Hsp70, and DnaJ with 78, 75, and 60 families, respectively. The type of HSP with fewer families was Hsp90 (8 families). Curiously, some large Hsp100 families are exclusive (or almost) of some genomes and do not contain any representant in our fungal transcripts. For instance, Amanita muscaria has a family with 28 proteins (and 1 from Piloderma croceum, OG0000025), Galerina marginata has one with 12 (and 1 from A. muscaria, OG0000064), another with 15 proteins (OG0000056), and Exidia glandulosa has one with eight proteins (OG0000072). Another interesting family is OG0000005, annotated as Hsp70; it has many proteins in 8 Basidiomycota genomes (32 in Serendipita vermifera, 24 in A. muscaria, 17 in G. marginate, and other Basidiomycota with less than 10) and only one protein in an Ascomycota genome (Fusarium oxysporum).

Transcript assignment to the Talaromyces genus

We identified which transcripts belong to the genus Talaromyces, as the taxonomic analysis with Kaiju pointed it as the most represented genus in Ascomycota. The percentage of Talaromyces transcripts were 9.7, 15.3, and 10.8% (total of 291, 662, and 371) for A. fourcroydes, A. sisalana, and hybrid 11648, respectively (Table S6). These results are in accordance with Kaiju. Higher expression transcripts were annotated as HSPs, or have functions related to protein repair, translation, or energetic metabolism. Curiously, in the three communities there were non-root-specific transcripts. In A. fourcroydes, there is an Hsp90 expressed in all tissues, whereas in A. sisalana and hybrid 11648 there are Hsp70 and a “translation elongation factor 1” expressed in all tissues. This suggests that Talaromyces might be present in all agave tissues in this region.

Discussion

In this study, we used bioinformatics tools to predict potential molecular functions of transcripts of fungal communities found on the transcriptomes of three agave cultivars. The percentage of fungal transcripts is consistent with a previous report that 21% of transcripts from stem samples in Eucalyptus grandis did not map to the genome of the host plant (Messal et al., 2019). Still, it is remarkable that there are more fungal root-specific transcripts than plant ones in the analyzed agave cultivars (Fig. 1A). These fungi could be epiphytic or endophytic. In the latter case, they could also be localized in the velamen region of roots. In A. sisalana, the velamen is composed of four layers of cells that have an irregular shape and absorbent hairs are found in groups on the roots, where associations with fungal hyphae may occur (Neto & Martins, 2012).

The three fungal communities we analyzed are taxonomically similar to each other, probably because all the plants were grown on the same field and climate conditions. This agrees with previous findings that, especially in agaves, geography matters more in fungal than in prokaryotic communities (Coleman-Derr et al., 2016). We noticed a large number of orphan transcripts in the orthologous analysis, which can indicate that these fungi are using different metabolisms in each plant.

Classified fungi mainly belong to the phyla Ascomycota and Basidiomycota and are present in similar proportions and numbers across the plants (Fig. 1B; Table S1). Most transcripts were classified as Ascomycota, as compared to Basidiomycota. Similarly, Citlali et al. (2018) reported, in four Agave species, a much higher proportion of Ascomycota fungi than from other phyla and a low abundance of arbuscular mycorrhizal fungi (AMF). In our data, only around 1.7% of conserved genes were assigned to AMF (Table S7), although AMF were already described in Agave species (Pimienta-Barrios, Zanudo-Hernandez & Lopez-Alcocer, 2009; Quinones-Aguilar et al., 2016; Montoya Martinez et al., 2019). To complement the results, we performed an rRNA identification analysis using the raw reads. The most abundant fungal genera in the roots were Parastagonospora, Talaromyces, and Cryptococcus. Indeed, our results pointed Talaromyces as the most abundant genus from Ascomycota (11.9% of Ascomycota transcripts) (Table S6). As our resolution would be too low to classify the transcripts into lower taxonomic ranks, we performed all other analyses at the phylum level.

As our main interest was finding whether the fungi could be interacting with the plants, we focused on finding molecular functions related to such interactions. To do so, we investigated enriched KEGG pathways and CDD domains (Fig. 2). One of the most frequent enriched KEGG pathways for all Ascomycota and A. sisalana Basidiomycota communities was the exosome, which plays roles in cell communication and nutrient delivery. Similarly, the membrane trafficking pathway was also enriched in the same groups. Associated with signaling pathways, transport of small molecules, and metabolic processes (Geisler, Murphy & Sze, 2013), it could be related to plant-microbiome interactions. All these enriched pathways related to transport are more frequent in Ascomycota than Basidiomycota and looking at the numbers of transcripts annotated as transporters (Table S4), Ascomycota indeed presents more transporters.

Analysis of secreted CAZymes suggests interactions between the fungi and the host plants, as endophytes must break the plant cell wall (PCW) to colonize the host. PCWs are mainly formed by cellulose, hemicellulose, lignin, and pectin, which form the first barrier against pathogens and other abiotic stresses (Benoit et al., 2015). Both Ascomycota and Basidiomycota in all three communities presented CAZymes related to degradation of PCWs, such as GH16, GH17, GH43, and AA9 (Table 4) suggesting that they could be colonizing the plant roots. Most of them were annotated as glycoside hydrolases, which is the family that has cellulolytic and hemicellulolytic enzymes. On the other hand, they also presented chitinases (GH18), related to the degradation of fungal cell walls. Secreted chitinases can be used as a defense mechanism against other fungi or plant pathogens, in the case of a symbiont relationship (Aranda-Martinez et al., 2016; Yang et al., 2019). More experiments are needed to address these aspects in agave roots. Although there is a lot of variety in the distribution and quantity of CAZymes in fungal genomes, symbiotic fungi on average have fewer CAZymes (Zhao et al., 2013).

Because agaves are well adapted to dry environments and the cultivars sampled in this study were going through a period of irregular precipitations, it was expected to find expressed transcripts related to abiotic resistance among the fungi. We found enriched protein domains and KEGG pathways related to chaperones and heat shock proteins, which are among the top expressed transcripts and the most frequent transporter families. HSPs have many different functions, many related to protein folding in some way. Particularly in fungi, they act in stress resistance, sporulation, sexual/asexual development, and virulence (Bui et al., 2016; Chatterjee & Tatu, 2017). Additionally, HSPs were also important drug targets in fungal-caused diseases (Lamoth, Juvvadi & Steinbach, 2016). One of their main functions, however, as the name suggests, is related to temperature stress. Our samples were collected at noon in a very dry region, and in similar environments the soil surface can reach over 40 °C (Nobel, 2010; Sattari, Dodangeh & Abraham, 2017). Therefore, it seems reasonable to find many HSPs as top expressed transcripts and within enriched categories, although more experiments are needed to test if they contribute to temperature stress responses.

To further investigate the different HSPs found in our datasets and to compare them to other fungi, we did an HSP orthologous analysis. The main type of HSP found in our agave fungal transcript datasets was small HSPs (Fig. 4), annotated as ACD (alpha-crystallin domain). ACD is a conserved domain through evolution, although the whole sequence of small HSPs varies (Kriehuber et al., 2010). Small HSPs can act in response to temperature stress; some of them were described as conferring tolerance to freezing (Pacheco et al., 2009) and heat shock (Haslbeck et al., 1999) in Saccharomyces cerevisiae. Also, small HSPs were described as the first defense against many stresses (Haslbeck & Vierling, 2015). In filamentous fungi, the number of copies of small HSPs in their genomes does not vary much (3–5 copies each) and they diverge a lot across fungal species (Wu et al., 2016). This could explain the difference between the number of proteins clustered in the other fungal genomes analyzed and our fungal transcripts (OG0000001), as maybe our samples presented closely related species grouped. However, there are more ACD-annotated proteins in Ascomycota, whereas the Basidiomycota fungal genomes present more copies of ACDs. This suggests that even if the Basidiomycota fungi present in the agave community also have many copies in their genomes, they are not expressing them.

To our knowledge, there are no works comparing HSPs across these fungal genomes. In this regard, one important protein family for the fungal genomes, both in Ascomycota and Basidiomycota, but not so present in our fungal transcripts, was DnaJ (Table S5). There are 60 families of DnaJ, being the third type of HSP more abundant in the whole dataset. DnaJ proteins confer stress protection, playing key roles in the cell death cycle and resistance to diseases (Liu & Whitham, 2013). The families with more proteins assigned to were Hsp100 (78 families) and Hsp70 (75 families). Hsp70 and Hsp100 both form a complex that acts in the disaggregation of other proteins, which could explain the similar number of transcripts belonging to these families in our agave datasets (OG0000002 and OG0000003). All HSPs were more uniformly distributed through Ascomycota genomes, but Hsp100 presented many exclusive or almost exclusive families in many Basidiomycota genomes. Thus, Hsp100 are seemingly important chaperones for some Basidiomycota.

Conclusions

In this study, we identified and characterized fungal transcripts found in the transcriptomes of three agave cultivars grown in the Brazilian semiarid without irrigation and under an irregular precipitation regime. We found more fungal root-specific transcripts than plant ones. These fungi mainly belong to two different phyla which are performing somewhat distinct functions, many related to interactions with the host plant and others related to drought resistance. Microbial communities can contribute to increasing the host plant’s resistance to many biotic and abiotic factors. Therefore, the current study underlines the importance of analyzing possible “contaminants” that may appear in transcriptome datasets, as valuable information might be present. In summary, our exploratory analysis of fungal communities provides a starting point to the prospection of potential microorganisms that could be exploited to generate improved agronomical characteristics in agave or other cultures.

Supplemental Information

Supplemental Information 1 Precipitation data from the city of Monteiro (Paraiba, Brazil)

The germplasm bank where the samples were collected is at a semi-arid climate area with non-calcic brown soil. Before sampling (July 6th, 2016) the precipitation regime was highly irregular. The last precipitation happened on May 31st, 2016.

Click here for additional data file.

Supplemental Information 2 Distribution of the number of HSPs in all Ascomycota and Basidiomycota genomes (Ensembl Fungi release 46)

To obtain all heat shock proteins in our fungal datasets and the fungal genomes (1,014 genomes), we searched with HMMER (E-value threshold of 0.001). We used the distribution pattern to select the Ascomycota and Basidiomycota genomes with more HSPs, which accounted for the top 10 genomes each.

Click here for additional data file.

Supplemental Information 3 Fungal transcripts expression data and annotation for each community

Expression values shown are the mean from the 3 biological replicates. “CAZy” sheet contains the result from the dbCAN2 software, “SignalP” the results from SignalP, and “TCDB” the BLAST results against the TCDB.

Click here for additional data file.

Supplemental Information 4 Orthologous analysis of fungal transcripts (nucleotide sequences)

Orthologous families from the three fungal communities, obtained with OrthoFinder.

Click here for additional data file.

Supplemental Information 5 Ribosomal RNA analysis for each read library

We used SortMeRNA with default parameters to filter the reads that had similarities to the 28S rRNA database. Then we filtered the reads belonging to fungi to assess the percentages of fungal reads in our datasets (“Percentages_28S” sheet). Finally, we listed the top 5 fungal species that had more abundant reads in each RNA-seq library (“28S_top5” sheet).

Click here for additional data file.

Supplemental Information 6 Numbers of transporters families annotated from TCDB in each fungal community

Classification was obtained with a BLASTp (E-value 1e−5) against the Transporter Classification Database (TCDB). We counted how many transcripts belonged to each TCDB family.

Click here for additional data file.

Supplemental Information 7 Heat shock protein (HSP) orthologous analysis between agave fungal communities datasets and other fungal genomes with many HSPs

The top 10 Ascomycota and top 10 Basidiomycota with more HSPs were compared to the agave fungal communities (Ascomycota and Basidiomycota for each plant community). Only HSP sequences were used in this analysis.

Click here for additional data file.

Supplemental Information 8 Talaromyces transcript assignment

We did a BLASTn (E-value 1e−10) of all Ascomycota transcripts against a set of concatenated genomes of Talaromyces available at Ensembl Fungi release 46. We selected only blast hits with alignment coverage above 70%.

Click here for additional data file.

Supplemental Information 9 Glomeromycota transcript assignment

We did a BLASTn (E-value 1e−10) of all fungal transcripts against a set of concatenated genomes of Glomeromycota available at Ensembl Fungi release 46. We selected only blast hits with alignment coverage above 70%.

Click here for additional data file.

Supplemental Information 10 Scripts used in the article

The script “classify_asco_or_basidio.pl” was used for the classification between Ascomycota and Basidiomycota. The files “asco_cds.fasta.id” and “basidio_cds.fasta.id” are used as input. The script “kegg_cdd_classification.md” was used for assigning KEGG and CDD functions to the Ascomycota and Basidiomycota transcripts.

Click here for additional data file.

Supplemental Information 11 FASTA files of transcript and protein sequences for the three fungal communities

Click here for additional data file.

Supplemental Information 12 Results of the bioinformatics pipeline for the classification of transcripts from Ascomycota and Basidiomycota

We used TPR and FPR values to compare parameters for classification of Ascomycota and Basidiomycota (see Methods section). The script is also available as a Supplemental File.

Click here for additional data file.

Additional Information and Declarations

Competing Interests

Author Contributions

Data Availability

The authors declare there are no competing interests.

Marina Püpke Marone conceived and designed the experiments, performed the experiments, analyzed the data, prepared figures and/or tables, authored or reviewed drafts of the paper, and approved the final draft.

Maria Fernanda Zaneli Campanari and Marcelo Falsarella Carazzolle conceived and designed the experiments, performed the experiments, analyzed the data, authored or reviewed drafts of the paper, and approved the final draft.

Fabio Trigo Raya and Gonçalo Amarante Guimarães Pereira conceived and designed the experiments, authored or reviewed drafts of the paper, and approved the final draft.

The following information was supplied regarding data availability:

The sequences are available at NCBI SRA: PRJNA746623. The assembled transcripts and protein sequences are available in the Supplemental File.

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
