# Peer review of "Fungal communities represent the majority of root-specific transcripts in the transcriptomes of Agave plants grown in semiarid regions"

_PeerJ, doi:10.7717/peerj.13252_

## Round 0.1 · original submission · Major Revisions

The manuscript received two detailed and constructive reviews. I agree with the reviewers that it would improve the manuscript if the number of libraries generated per tissue sample and the RNA sample was processed were provided. I also agree that the text needs revision for format and style. I made dozens of edits in to the first 300 lines (see attached PDF). Please use these edits as a template for the entire manuscript.

Other comments

Recent literature was ignored, including Flores-Núñez et al. (2020 10:3380).

Make Figure 2 full page.

Reviewer 1 ·

Basic reporting

The article is mostly well written. However, some sentences are too long and should be shortened for clarity. Some examples are:
Lines 118-121, 184-187, 197-199, etc.

Some references related to the fungal part of the microbiome associated with Agave species and their effects are missing, such as:
doi: 10.1111/1462-2920.15395.
https://doi.org/10.1016/B978-0-08-102268-9.00012-4
10.1111/j.1469-8137.1992.tb00092.x
https://doi.org/10.1023/A:1024046925472
http://www.scielo.org.mx/scielo.php?script=sci_abstract&pid=S0187-71512009000400005&lng=es&nrm=iso&tlng=es
http://www.scielo.cl/scielo.php?script=sci_abstract&pid=S0718-95162016000400016&lng=es&nrm=iso&tlng=en
http://www.ajol.info/index.php/ajb/article/view/95729

Raw transcriptomic reads have been shared, as two papers based on the RNA-Seq analyses have been published already (plant transcripts and viruses). I identified however, that only 17 libraries are associated with this accession number, although I expected 27 or 18 (3 Agave cultivars, 3 plant compartments (roots, stem and leaves) and 3/2 biological replicas).

Authors should include quantitative data about the size of the Agave plants sampled, the properties of the soils (if possible), and also about the precipitation patterns. They argue these plants have suffered irregular precipitation periods, but no evidence of this is provided.

Experimental design

The research question was well defined. However, the methods are not fully and clearly described to allow replication.

For instance, how total RNA was obtained from collected plant samples is not fully described. Did authors used a kit to deplete ribosomal RNAs? Did they use PolyA tails to recover only eukaryotic mRNA sequences? If total RNA was sequenced, can authors also provide a general view of the prokaryotic mRNAs in the leaves, roots and stems of the three Agave cultivars investigated too? how many.. I realized that authors have already published the analyses on the plant transcripts and also on the viruses! That is great! However, this paper on fungal transcripts is insufficiently documented to be clearly followed.

To my knowledge and expertise, ribosomal RNA depletion efficiency is always below 100% and thus, rRNAs are normally sequenced and can be used for taxonomic identification. Can authors recover and report fungal rRNAs sequences? That would link their functional annotations to more specific lineages. This information would also allow the mapping of the fungal transcripts to "specific" fungal genomes recovered from these agaves.

One of the major aspects missing is the analysis of the distribution and identity of fungal transcripts in the three plant niches or compartments (root, stem and leaves) for each Agave species. Why only root associated fungal transcripts are described and discussed? I strongly suggest authors to give an overview of the differences between plant niches and cultivars.

The methods used for the annotation of the fungal transcripts are not sufficiently documented.

Ideally, authors should make the in-house transcripts publicly available in a platform such as github.

I could not find clear references to the fungal genomes used in this study outside "Ascomycota" and "Basidiomycota" (nor in the main Manuscript nor in the Supplemental Material). Did authors fully neglected AMF or other Mucoromycota? to date there are at least 10 genome assemblies of AMF in public databases, and they are important fungal symbionts of plants, including agaves.

Validity of the findings

One of the major points that need revision is the lack of analyses of the different plant compartments sampled, that is the roots, stem and leaves of the 3 Agave cultivars. Authors only discussed and presented detailed data about fungal root transcripts, but an overview of the fungal transcripts in the other plant niches is missing.

The findings mostly describe the expression of certain fungal genes in the plant samples, but these expression can not be directly linked to mechanisms or causality of drought resistance and/or pathogen defense. So, authors should revise the discussion of their results to reflect this limitation. Further experiments or analyses would be needed for that.

In general, authors could compare in more detail their results to the fungal phylogenetic profiles generated already for other Agave species (line 393-396). Indeed from these reports, no differences in fungal diversity associated with the roots of cultivated and wild Agave species was observed. Contrary to the claim authors make in line 396-400.

·

Basic reporting

Language
The manuscript is written in a way that one is able to follow the author’s intentions. However, the text should be revised for stringency, as there are numerous unnecessary words. Struggling with similar issues, I have started to look into options such as “Grammarly”, “Slick Write” or “Hemmingway Editor” for revising text. Potentially, these programs (all of them have a free version and there are plenty of other options) could help with avoiding lengthy sentences by assisting with punctuation and also with (limited) grammar suggestions.

Additionally, I would suggest to the authors to critically revise the structure of some paragraphs. In the Introduction and Discussion, it is sometimes difficult to follow the common thread within a paragraph and between sentences (e.g., line 60 ff.: This sentence describes what endophytes & epiphytes are and that they can play a crucial role in plant performance. The following sentence (line 64 ff.) describes the ambiguous relationship between mutualism and parasitism. Other examples: line 93 & line 95). Stating clearly defined research questions and following them could help structuring especially the Discussion. In the Methods section, the reader would profit by understanding at the beginning of the paragraph, why a specific analysis was performed. In several instances, this only becomes clear in the middle of the paragraph (e.g., line 155 ff. - Taxonomic classification, line 176 ff. – Orthologous gene analysis).

Intro & background
The introduction gives background on the diversity of plant-associated microbes, some factors structuring these microbial communities, and the study system agave (but see comments on stringency and common thread). It would be helpful to also find information on using metatranscriptomics to gain insight into microbe-plant interactions on a community level, as well as the difficulties faced by such approaches.

Figures
The photos in Fig. 1 should be at a higher resolution. Potentially they could also be bigger.
Figure 1 A: Only plant root-specific transcripts > Root-specific plant transcripts
Only fungal root-specific transcripts > Fungal transcripts in roots
Figure 1 B & 2: There seems to be enough space in the figures in order to use unabbreviated cultivar names. It would be much easier for the reader to see which sample belongs to which cultivar
Figure 4: Provide only information for data shown in the figure. Also label the scale in the figure with “Number of proteins”

Legends and captions for the Supplemental Material are missing, making it in some cases impossible to use (e.g., Table S4)

Raw data
Is available in NCBI SRA.

Experimental design

Research question
The research questions should be stated in a more explicit way at the end of the Introduction. Potentially, they could be numbered. This would also help to structure the Discussion.

Methods
Methodology follows standard procedures and is described in a way that makes it possible to follow the approach (but see specific comments for methods where I think more information is needed).

Validity of the findings

As stated above, all underlying data have been provided.

The conclusions of the study are drawn on the taxonomic level of the divisions Ascomycota and Basidiomycota. At this level the differences in molecular function in my opinion are not overly meaningful, as both groups are very divergent when it comes to life styles. The molecular functions amongst members of both groups could be more divergent than between the two groups. It would be more helpful to discuss the findings on lower taxonomic ranks and it is surprising that the authors refrain from this. They clearly have performed the taxonomic assignments at lower levels (e.g., Figure 4; Kaiju and blastx analyses should also give better taxonomic resolution). Just looking at Fig. 4 shows that there are several fungal genera that are potentially mycorrhizal partners (Amanita, Pisolithus, Rhizopogon, Tulasnella), endophytes (Serendipita),pathogens (Fusarium, Verticillium) or yeasts (Zygosaccharomyces).

Related to the previous comment might also be the fact that many conclusions and argumentations in the Discussion are not grounded on the results of the manuscript. Often the paragraphs become very speculative (e.g., line 435ff.: this paragraph describes the sugar metabolism, but in the whole paragraph the results of the study are only mentioned in one sentence (line 442), without going into details of the findings and what they could potentially mean).

Additional comments

Specific comments
Line 58: … and investigating them … -> … and investigating these …
Line 64: I would suggest using another reference than Faust & Raes 2012 to provide information on the diversity and function of plant-associated microorganisms. Faust & Raes 2012 is an article about microbial networks and models. There are plenty of reviews about plant-associated microorganisms (e.g., Vorholt 2012 Nature Reviews Microbiology; Fitzpatrick et al. 2020 Annual Review of Microbiology; Trivedi et al. 2020 Nature Reviews Microbiology)
Line 66: … plant biome … > … plant microbiome …
Line 71: … drought mechanisms, commercial uses, … bioenergy production … > These properties of agaves should be described in separate sentences, as drought mechanisms are physiological traits of the plant, whereas the other properties are the value that humans gain from the plant.
Line 90: In this context, for agave plants, we believe, tolerance to heat and drought stress is in part … > In agave plants tolerance to heat and drought partly might be due to …. This hypothesis needs more justification in the form of references. It should not be difficult to find these, as there are plenty of studies showing the influence of root-associated fungi on plant performance.
Line 105: … a cryptic species in A. niger (Duarte et al., 2018). > Please delete this part, as it sounds as if A. welwitschiae is not a species on its own.
Line 117: … these fungi communities … > … these fungal communities … This should be checked throughout the manuscript. It occurs several times.
Line 134: … at SRA (accession … > … at the NCBI SRA (accession …
Line 137: “The complete pipeline“ can unfortunately only be seen by readers that have a subscription to the journal “Industrial Crops and Products”. Please provide a sufficient description of the pipeline
Line 143: It is great to see that so much effort went into correctly identifying fungal and plant transcripts. However, I wonder why in subsequent taxonomic annotation the result of the Kaiju analysis was used and not the blastx/uniref90 approach. Additionally, no information is provided in the results how well the latter approach performed.
Line 144: Please provide more information (especially on parameter settings) on the Kaiju classification.
Line 155: Taxonomic classification. Please describe at the beginning of this paragraph in which way the two phyla were compared? This seems essential for the whole paragraph and is only described from line 169 onwards.
Line 174: The text in Methods S2 is nearly identical to the Methods text in the manuscript. I think differences could easily be incorporated into the main text. The results shown in Methods S2 could go into the Supplemental Material though (with a proper legend).
Line 209: Please explain why only the top 10 genomes of Ascomycota & Basidiomycota were used for HSP orthology analysis.
Line 210: If only the top 10 genomes from Ensembl plus three datasets from this study were selected, how does this add up to 26 datasets? Although the Methods section say 3 cultivars times three samples, but the results never show these 9 samples? Therefore, I was under the impression the individual samples for each cultivar were pooled.
Line 214 ff.: General sequence/assembly statistics at the beginning of the Results would be beneficial. Potentially, a table showing this would be good
Line 217: These plants were in … > I think the sentence either belongs into the Introduction or the Discussion, as it is not a result. Additionally, more justification (i.e., relevant literature) is needed for assuming that the fungi contribute to the phenotype of the plant.
Line 221: Are these transcripts now root-specific or transcripts found in all tissues of the different cultivars?
Line 224: Expression analyses need to be described in the Methods section.
Line 232: It is unclear what analysis is described. I think what is meant is a comparison of orthologous genes/transcripts based on nucleotide similarity.
Line 240: … probably indicating that > This should be moved to the Discussion, as it is already an interpretation of the results.
Line 348: … annotated as ACD. > What does the abbreviation ACD stand for. As it is highlighted in the text, the Discussion (line 465) should revolve around what ACD is and how it could be important.
Line 368: I would disagree that the study characterized the molecular functions of the transcripts of the fungal communities, as this entails detailed molecular studies in a laboratory. The presented work used bioinformatic tools to predict potential molecular functions.
Line 371: Why was the percentage of fungal transcripts a surprise? Already in the Introduction it is mentioned that in a previous companion study the authors found 10% of the transcripts to be fungal.
Line 376ff.: Although agaves contain a velamen as do many orchids, it might not have the same function. Especially, as the orchid in the citation (Deepthi & Ray 2018) is an epiphytic orchid while the roots of agaves are in the soil. It is also curious why the emphasis of this paragraph is on the velamen, as it is not functionally related to CAM. There are CAM plants without velamen and plants with velamen that are not CAM plants.
Line 385: Please explain how the intention to sequence the plant explains the low expression values of the fungus? I would argue that the low expression values of the fungi are related to the low fungal biomass in the plant compared to plant biomass.
Line 401: In which way where the communities very similar? Functionally, taxonomically, etc.)
Line 403: … previous findings that fungal communities depend vastly on the soil (Lee & Hawkes, 2020) … > This is exactly what the study by Lee & Hawks does not find. Their study indicates that there is little overlap between leaf and root fungal communities (their Fig. 1 A). Only 1.2% of ASVs overlap between the two tissues.
Line 456: Particularly … > Please revise the sentence as it is not logically coherent. The first part of the sentence deals with physiological properties of HSP in fungi, whereas the second part deals with how HSP can be used by humans to treat diseases.
Line 460: At what depths were the samples collected? This would be important to put the statement “soil surface can reach over 40°C” into context. Additionally, is the Nobel 2010 (Desert wisdom/Agaves and Cacti: CO2, water, climate change) the right reference for the surface temperatures at the sampling site?

---

## Round 0.2 · Minor Revisions

We only received one review of this submission (version 1), so I have decided to proceed with the recommendation of minor revision. I expect I can make a decision, without a further round of review, if these comments are addressed.

Regards,

Michael

·

Basic reporting

I think the authors have addressed most of the concerns raised in the previous round of reviews and the manuscript has improved significantly. Still, there are several aspects, which I think should be addressed before publication can be considered.

General comments & suggestions:
Please check the tenses throughout the manuscript, as these often switch between present and past tense even within individual paragraphs.

It would be helpful for the reader if the authors could be consistent with the naming of taxonomic groups. For instance, throughout the text Basidiomycota (e.g., line 343), Basidiomycetes (e.g., line 353) or basidiomycetes (e.g., line 352) can be found. These names do not necessarily represent the same taxonomic group and could be confusing to the reader.

As suggested by reviewer 1 the authors included now analyses of rRNAs for taxonomic assessments of the fungal communities. There was some interesting information obtained on potential fungi being present in these communities, for which ecological information is available. For instance, there are several yeast taxa (Cryptococcus, Saccharomyces, Malassezia), one lichen species (Verrucaria), but also a potential parasite (Erysiphe). However, this information is only sparsely used in the Discussion to obtain a taxonomic context for the transcriptomic results. Obviously, caution is needed to correlate the rRNA results with the transcriptomic results, nevertheless I think it would add the results into perspective.

Specific comments & suggestions

Line 49: Plants hosts many microorganisms and investigating their metabolism can … > Plants hosts many microorganisms and investigating the metabolisms of these can …
Line 68: … than other aspects still under investigation. > … than other aspects.
Line 86: … three papers … > … three studies
Line 107: … (2) identified their main phyla … > … (2) assessed their taxonomic affiliations …
Line 108: … including transporters …. > … focusing on transporters …
Line 109: … (4) investigated their heat shock proteins, as these appeared consistently as most expressed transcripts. > … (4) investigated potential heat shock proteins.
Line 110: Around 12% of the full transcriptomes … > I think these sentences should be removed, as they present Results and not Introduction.
Line 124: We have harvested … > We harvested … Please scrutinize the manuscript carefully as many instances occur where “have” can be removed.
Line 128: … 1ug … > … 1 µg …
Line 140: … we have done … > … we performed … (also in line 158)
Line 307: … it is found … > … is found …
Line 308/9 … also a domain present in HSPs. > … also an HSP domain.
Line 310: Regarding central metabolism … > Regarding primary metabolism
Line 350: Please explain in the text what is meant by “plant profile”.
Line 365: … Basidiomycota, considering the total for the three communities. > … … Basidiomycota in the three communities.
Line 389 ff.: Please explain to the reader how this part of the study (i.e., HSPs in fungal genomes) is relevant to the study. These are interesting results, but they seem to justify a separate manuscript.
Line 418: … experiments, we believe that … > … experiments, often …
Line 429: We understand that our dataset … metatranscriptome, but the number … transcripts is sufficient … > Although, our dataset … metatranscriptome, the number … transcripts seems sufficient …
Line 433: Nevertheless, … > Please revise this part, as it is not possible to see the connection to the previous sentence.
Line 437: … the plants are on the same field and climate conditions. > … the plants were grown on the same field.
Line 449: We too were not able … > This sentence seems to contradict itself, as 1.7% of the transcripts belonged to AMF lineages, but not AMF were identified.
Line 453: Please revise the sentence. Improving plant performance might be the main goal of this research route and the study certainly could give hints as to which genes/transcripts could be potential candidates for further research. However, due to the study being exploratory and not experimentally it is impossible within its frame to pinpoint actual function of certain transcripts.
Line 457: … nutrients delivery. > … nutrient delivery.
Line 470: On the other hand … > Please revise the sentence. It is not clear as to whether the presence of chitinases could be a defense mechanism of the fungi against mycoparasitic fungi (in which case it is not clear how this relates to the research topic) or whether it is a defense mechanisms that the authors assume to be a mechanism that these endophytic fungi use in order to defend their host plant.
Line 476: … symbiotic fungi can have fewer … > … symbiotic fungi on average have fewer …
Line 527: … study underlies the importance … > … study underlines the importance …
Line 529: In summary, … > Like for the sentence in line 453, I think this statement needs to be revised. In my opinion the present study can only hint to

Experimental design

I have no further additions to the first review

Validity of the findings

I have no further additions to the first review

---

## Round 0.3 · Minor Revisions

I think the reviewers comments are adequately addressed but the manuscript still needs edits for style. The text is wordy and includes words like "anyhow" and phrases like "bearing in mind" that add little or no content. I suggest the following changes a examples. Please follow these examples throughout. That is, do not just make these changes. Try to write more succinctly throughout.

Regards,

Michael

Line 28. Revise to “These plants host many fungi that form beneficial relationships with agave and have biotechnological applications. To identify and characterize these fungi, we sequenced transcripts….”
Line 43. Why is “membrane trafficking” italicized?
Line 49. Replace “microorganisms and investigating the metabolisms of these can be a starting point for the development of biotechnological tools. Microbiomes can have crucial…” with “microorganisms that have crucial…”
Line 68. Replace “In this context, more recently some works focused on the..” with “Some recent research has focused on the…”
Line 89. Replace “In summary, meta-omics approaches allow the discovery of plant-microbe interactions, revealing the microorganisms present in a community and their functions. Metatranscriptomics has advantages over the use of sole metagenomics, because apart from taxonomy and gene sequences it is possible to obtain the set of functional genes that are being expressed in the community under the conditions studied” with "Metatranscriptomics can identify genes expressed in microbiomes.”
Line 112. Delete “The” in “The samples” and use this determiner sparingly throughout.
Line 261. Replace 1.49 with 1.5
Line 290. Why are phrases like “chaperones and folding catalysts” italicized?
Line 321. Delete phrase “it was described that”
Line 343. Delete phrase “It is possible to notice again the majority of GHs.”
Line 344. Two “others” doesn’t make sense. Delete “other” in “other types”
Line 346. Replace “we can notice that Basidiomycota has more secreted” with “Basidiomycota secreted more”
Line 356. Delete “Furthermore, there are not many studies focusing on a broad comparison between fungal HSPs. In our data,”
Line 410 - 416. Delete “namely A. fourcroydes, A. sisalana, and hybrid 11648” and replace “The percentage of fungal transcripts in our plant transcriptome dataset …plant genome.” With “The percentage of fungal transcripts is consistent with a previous report that 21% of transcripts read from stem samples in Eucalyptus grandis did not map to the genome of the host plant..”
Line 423. Delete “Anyhow, fungal…dry environments.”
Line 437. Delete “The” in “The classified” (see above)
Line 448. Delete “We too were…(Table S12).” And revise line 444 to “Most transcripts read from agave samples classified as Ascomycota, as compared to Basidiomycota, and only around 1.7% of conserved genes were assigned to AMF (Table S12).”
Line 462. Replace “The analysis of secreted CAZymes also provides a hint of possible interactions …” with “Analysis of transcripts related to secreted CAZymes suggests interactions…”
Line 474. Delete “it was demonstrated that”
Line 476. Replace “Bearing in mind that agaves are plants adapted to dry environments and the ones analyzed in this study were going through a period of irregular precipitations, it was…” with “Agaves are well adapted to dry environments and the cultivars sampled in this study were experiencing a drought, it was..”
Line 485. Delete “it has been described that” and “it was shown that” (line 498).
Line 509. Delete “are known molecules that”

---

## Round 0.4 · Minor Revisions

The manuscript appears acceptable but before I can make that decision, please address the following suggestions. Note, format of references in not required for acceptance by this journal but I think it is good form. Please double check references.

Line 26. Replace “feedstock to bioenergy” with “feedstock for bioenergy”
Line 28. Something is missing here. Replace “… production in marginal areas. Plants host many microorganisms that can be used for biotechnological purposes.” With “… production, Fungi associated with these plants may contribute to the ability of their hosts to thrive in marginal areas. These fungi may have biotechnological applications.”
Line 57. Replace “agaves are also interesting to be used as a biomass feedstock to bioenergy” with “…agaves also show promise as feedstock for bioenergy..”
Line 74. Delete “and not only due to physiological mechanisms of the plant itself.”
Line 93. Replace “The region” with “This region”
Line 118, 150, 241, 308 and 309, 311, 312. 316. Delete “The”
Line 309 Insert “and” (….11648 and…)
Line 407. Replace “this is in agreement with” with “this agrees with”
Line 537 Why all caps manuscript title? Also, italicize genus and species (see 550, 557, 634…)
Line 540. Again, italicize genus species. Also, cap first letter in journal titles (PloS Neg..). See also 589, 613, 617, 644….
Line 543. This reference is missing the book title.
Line 554. Only cap first letter of proper nouns in manuscript titles (see also 559. 562, 583, 591).
Line 560. Add page numbers.

---

## Round 0.5 · accepted · Accept

I appreciate your patience.

Regards,

Michael